# Exploring the Benefits of Probiotics in Gut Inflammation and Diarrhea—From an Antioxidant Perspective

**DOI:** 10.3390/antiox12071342

**Published:** 2023-06-26

**Authors:** Qihui Li, Tenghui Zheng, Hanting Ding, Jiaming Chen, Baofeng Li, Qianzi Zhang, Siwang Yang, Shihai Zhang, Wutai Guan

**Affiliations:** 1Guangdong Province Key Laboratory of Animal Nutrition Control, College of Animal Science, South China Agricultural University, Guangzhou 510642, China; 3407lqh@stu.scau.edu.cn (Q.L.); ht.d@stu.scau.edu.cn (H.D.);; 2College of Animal Science and National Engineering Research Center for Breeding Swine Industry, South China Agricultural University, Guangzhou 510642, China; 3Guangdong Laboratory for Lingnan Modern Agriculture, South China Agricultural University, Guangzhou 510642, China

**Keywords:** inflammatory bowel disease, oxidative stress, probiotics, weaned piglets

## Abstract

Inflammatory bowel disease (IBD), characterized by an abnormal immune response, includes two distinct types: Crohn’s disease (CD) and ulcerative colitis (UC). Extensive research has revealed that the pathogeny of IBD encompasses genetic factors, environmental factors, immune dysfunction, dysbiosis, and lifestyle choices. Furthermore, patients with IBD exhibit both local and systemic oxidative damage caused by the excessive presence of reactive oxygen species. This oxidative damage exacerbates immune response imbalances, intestinal mucosal damage, and dysbiosis in IBD patients. Meanwhile, the weaning period represents a crucial phase for pigs, during which they experience pronounced intestinal immune and inflammatory responses, leading to severe diarrhea and increased mortality rates. Pigs are highly similar to humans in terms of physiology and anatomy, making them a potential choice for simulating human IBD. Although the exact mechanism behind IBD and post-weaning diarrhea remains unclear, the oxidative damage, in its progression and pathogenesis, is well acknowledged. Besides conventional anti-inflammatory drugs, certain probiotics, particularly *Lactobacillus* and *Bifidobacteria* strains, have been found to possess antioxidant properties. These include the scavenging of reactive oxygen species, chelating metal ions to inhibit the Fenton reaction, and the regulation of host antioxidant enzymes. Consequently, numerous studies in the last two decades have committed to exploring the role of probiotics in alleviating IBD. Here, we sequentially discuss the oxidative damage in IBD and post-weaning diarrhea pathogenesis, the negative consequences of oxidative stress on IBD, the effectiveness of probiotics in IBD treatment, the application of probiotics in weaned piglets, and the potential antioxidant mechanisms of probiotics.

## 1. Introduce

Inflammatory bowel disease (IBD) is an incurable chronic inflammatory gastrointestinal disease that primarily includes Crohn’s disease (CD) and ulcerative colitis (UC). The global prevalence of IBD has rapidly increased in recent years, with approximately 6.8 million people suffering from IBD in 2022 [1]. IBD types are typically distinguished by the location of inflammation and the histopathological characteristics of the gastrointestinal tract. Clinically, CD can occur in any region of the gastrointestinal tract, including the ileum and colon, characterized by transmural inflammation. On the other hand, UC specifically appears in the colon and rectum, with inflammation limited to the mucosa [2]. As an inappropriate immune response, the causes of IBD are considered multifaceted, involving genetic predisposition, environmental factors (Western diet, poor sanitation, and smoking), damage to intestinal epithelial integrity, and dysbiosis in the gut microbiome [3]. The exact underlying mechanisms of IBD remain unknown; however, accumulated data from animal experimental models and clinical studies suggest that oxidative stress (OS) signaling occupies a dominant position in the pathogeny of IBD. In brief, OS leads to damage to the gastrointestinal mucosal layer and dysbiosis, which are important features of IBD patients. This, in turn, stimulates immune responses and triggers IBD [4]. Therefore, relieving systemic oxidative stress becomes a crucial goal in treating IBD. Currently, the supplementation of reactive oxygen species (ROS) production inhibitors, corticosteroids, aminosalicylates, and substances that stimulate endogenous antioxidant enzymes have gradually emerged as complementary and alternative therapies for IBD treatment [5]. In addition, probiotics are being developed as therapeutic strategies for IBD. Studies have shown that probiotics, particularly *Lactobacilli* and *Bifidobacteria*, offer benefits for IBD patients by improving intestinal microecology, protecting intestinal mucosal barrier integrity, and modulating immune responses [6]. These effects are associated with their ability to scavenge ROS, chelate metals, and regulate the levels of host antioxidant enzymes [7]. Moreover, evidence from clinical studies suggests that the transplantation of healthy donor-derived microbiota to IBD patients promotes the recovery of their gut microbiota and the resolution of inflammation [8]. Therefore, approaches such as gut microbiota transplantation or the oral administration of probiotics hold potential as therapeutic interventions with which to alleviate clinical symptoms in IBD patients.

In pork production, weaning and feed transition often lead to intestinal barrier damage, intestinal villus atrophy, and an overload of proinflammatory factors (TNF-α, IL-6), resulting in diarrhea, decreased feed intake, and compromised growth [9]. Post-weaning diarrhea (PWD), caused by intestinal inflammation and oxidative stress, contributes to significant economic losses. Traditionally, antibiotics have been used as a means to alleviate diarrhea and promote growth [10]; however, due to the rise in antibiotic resistance among intestinal pathogens and concerns about drug residues, antibiotics are gradually being banned in many countries. Currently, certain probiotics have been suggested as alternatives to antibiotics in weaned piglets, exerting anti-inflammatory and antioxidant effects, modulating the microbiome, enhancing intestinal epithelial barrier function, and alleviating diarrhea [11,12].

In clinical research, difficulty in sampling, environmental limitations, and ethics often hinder direct research on human diseases. For a long time, small rodents have been important model animals for basic medical research, making significant contributions to the understanding of the pathogenesis and treatment of human diseases; however, pigs are more similar to humans in terms of physiology and anatomy than rodents, making them a potential choice for simulating human diseases [13]. At present, there are pig models for cardiovascular diseases, metabolic disorders, and neurological diseases, which provide considerable support for the analysis and treatment of human diseases [14]. Therefore, we take the intestinal inflammation of weaned piglets as an example with which to discuss the following points sequentially: the relationship between oxidative stress and IBD, the potential of probiotics in IBD treatment, the application of probiotics in weaned piglets, and the possible mechanisms of probiotics in IBD treatment.

We conducted a thorough search using PubMed, Medline, and Web of Science databases from 2010 to 2023, and found a total of 41 papers to include in this review. These papers included in vitro cell tests, small rodents (mainly induced with dextran sodium sulphate or 2,4,6-trinitrobenzenesulfonic acid), and piglets (induced with weaning stress) as IBD models. In addition, we also searched for double-blind, placebo-controlled trials of adults and children with active or quiescent CD or UC within the past 20 years. This review provides valuable insights into the potential of probiotics, particularly lactobacillus, in alleviating IBD symptoms.

## 2. Oxidative Stress in Inflammatory Bowel Disease

OS is an imbalance between oxidants and antioxidants, with reactive oxygen species (ROS) being the most common highly reactive molecules in organisms. ROS, including superoxide (O_2_·^−^), peroxy radical (RO_2_·), hydroxyl radical (HO·), and hydroperoxy radical (HO_2_·), are natural byproducts of metabolism [15]. They are primarily produced by organelles such as the endoplasmic reticulum, mitochondria, and peroxisome, as well as by enzymes, such as peroxidase, NADPH oxidase, xanthine oxidase, lipoxygenase, glucose oxidase, and epoxidase [16]. Antioxidant systems, on the other hand, consist of enzymatic and nonenzymatic defenses. Enzymatic defenses, including catalase, superoxide dismutase (SOD), and glutathione peroxidase (GP-x), are present in all cells. Nonenzymatic defenses typically involve substances such as glutathione, ascorbic acid, vitamin E, C, A, and metal elements (zinc, copper, manganese, and iron) [2]. At homeostatic levels, ROS have numerous physiological functions, such as cell signal transmission, growth, differentiation, apoptosis, and inflammation; however, under OS, overloaded ROS would damage cell biomacromolecules, especially membrane lipids, DNA, and proteins [2].

### 2.1. Oxidative Stress Is the Trigger of IBD

Currently, OS is receiving increasing attention as a potential etiology or trigger of IBD. Numerous studies have compared the levels of OS markers between healthy individuals and IBD patients. The results have shown that the levels of antioxidant enzymes (including PON1, SOD, CAT, and GP-x) and nonenzymatic antioxidant substances (vitamins A, C, E, and β-carotene) are higher in healthy individuals compared to IBD patients [4,17]. Additionally, the most commonly evaluated index is the total antioxidant status (TAS) or total antioxidant capacity (TAC), which reflects the overall antioxidant capacity of an individual [18]. The study found that the TAS/TAC in the serum or plasma of adult patients with IBD (including CD and UC) uniformly decreased [3]. Conversely, pro-oxidases/agents, such as MPO, NO, spermine oxidase, COX2, NOX2, and NOS2, were all raised in the intestinal mucosa and serum of IBD patients [19]. Moreover, the concentrations of lipid peroxidation products (4-hydroxynonenal and malondialdehyde), DNA oxidation products (8-OHdG), and oxidative protein products (hydroxylated or carbonylated proteins) with proinflammatory properties have been shown to be positively correlated with the severity of IBD [20,21,22]. Additionally, oxidative-damage-induced DNA strand breakage, pyrimidine/purine loss, or abnormal pyrimidine and purine modification are considered key factors in the occurrence of IBD [3,23]. Similarly, another index, the “Oxidative Stress Index”, obtained by dividing the total oxidative capacity by the total antioxidant status, is significantly higher in IBD patients compared to the healthy population [3].

OS is closely associated with the main pathological feature of IBD, which is inflammation. Studies have found a positive correlation between indices of OS and levels of C-reactive protein, an inflammatory marker in CD patients [24]. Similarly, clinical evidence suggests that plasma free thiols, the main substrates of ROS, are inversely correlated with inflammatory biomarkers [24,25]. Mechanistically, redox signaling stimulates NF-κB signaling, which is intimately involved in the upregulation of inflammatory cytokines (IL-1, IL-8) and inflammatory cell infiltration. During the immune response, polymorphonuclear leukocytes and monocytes infiltrate massively into the injured intestinal mucosa, stimulating the ROS/RN-generating system to increase oxidative stress [24] (Figure 1).

### 2.2. Oxidative Stress Leads to Intestinal Dysbiosis

Notably, intestinal dysbiosis, marked by microbial diversity and a decrease in beneficial bacteria alongside an increase in pathogenic bacteria, is another prominent feature of IBD patients [26]. Accumulating evidence suggests that disharmony between the intestinal flora and the immune response of the intestinal mucosa occupies a central role in IBD pathogenesis [27]. Some opportunistic pathogenic bacteria (*E. coli* and *Helicobacter pylor*) have been identified as the main sources of intestinal redox signals, which directly or indirectly (stimulate neutrophils) produce ROS, leading to the development of IBD exacerbations [28,29]. Additionally, the differential structure of the gut microbiota in patients with IBD compared to healthy individuals has been extensively studied [30]. The abundance of beneficial bacteria normally present in the gut of healthy individuals, such as *Bacteroidetes* and *Firmicutes*, is significantly reduced in patients with IBD, while harmful populations, such as *Proteobacteria* and *Actinobacteria*, are increased [31]. Among the beneficial bacteria, *Faecalibacterium*, producing butyrate, with anti-inflammatory effects, is one of the most abundant species in the human gut [26]; however, the abundance of *Faecalibacterium prausnitzii* is decreased in the gut of IBD patients [32]. Similarly, levels of *Roseburia* spp., another butyrate-producing bacteria, are remarkably lower in populations at a high genetic risk for IBD [26]. Additionally, the abundance of *Bifidobacterium* is also decreased. In terms of pathogenic bacteria, the relative abundance of *Proteobacteria* (mainly *Escherichia coli)* and proinflammatory properties (*Escherichia* and *Fusobacterium*) is higher in IBD patients [33]. Similarly, a TNBS-induced murine model found raised populations of *E. coli* as well as *Clostridium* spp. and reduced populations of *Bifidobacterium* and *Lactobacillus* [34]. Dysbiosis leads to damage of intestinal mucosal integrity, which causes opportunistic bacteria to invade the mucosa, leading to inflammatory cascades [35].

Additionally, oxidative stress and inflammation caused by ROS overload are tightly intertwined, leading to intestinal mucosal barrier damage in IBD patients. This, in turn, increases mucosal permeability, allowing pathogen invasion, which further stimulates proinflammatory factor and ROS production, creating a vicious cycle [36].

Currently, the focus of IBD therapy is on reducing inflammation, and mainstream drugs include combinations of immunosuppressive and anti-inflammatory agents, such as anti-TNF-α antibodies and corticosteroids [37]. Furthermore, many antioxidant therapies, such as ROS production inhibitors, dietary interventions, and antioxidants, are being investigated as auxiliary therapies for IBD, exhibiting promising results [5]. Since gut microbiota interfere with both local and systemic immune responses, and their dynamic changes markedly depending on environmental factors and IBD treatment [38], supplementing IBD patients with probiotics that have antioxidant capacity might be a potential new therapy.

## 3. Probiotics in the Treatment of IBD

The advantages of probiotics in the treatment of IBD have been extensively studied in recent decades. The use of probiotics in patients with IBD has increased by 50% in recent years. Accumulating evidence suggests that certain probiotic strains are beneficial for the treatment and prevention of IBD, both in animal models and humans. The most commonly used probiotics are *Lactobacilli* and *Bifidobacteria*, which have been reported to improve the total antioxidant status of IBD patients. This improvement may be attributed to their ability to scavenge reactive oxygen species (ROS), chelate metals, stimulate host antioxidant enzyme levels (SOD, CAT, and GP-X), and modulate the gut flora [39,40,41].

### 3.1. Effect of Probiotics on Alleviating UC

The colon harbors the highest concentration of microbes in the human body. Several probiotics that normalize the composition of the colonic microbiome have shown benefits for patients with ulcerative colitis (UC). Currently, dextran sulfate sodium (DSS) and 2,4,6-trinitrobenzene sulfonic acid (TNBS) are commonly used to induce experimental models of UC and CD, respectively. In mouse model of DSS-induced colitis, *Bifidobacterium lactis A6*, *Bifidobacterium longum. infantis BB-02*, and *Bifidobacterium animalis lactis BB12* have been shown to inhibit OS, reduce colonic inflammation, and improve intestinal permeability [42]. Similarly, *Lactobacillus plantarum 2142* inhibited oxidative-stress-induced proinflammatory cytokine overexpression in the IPEC-J2 cell line [43]. In human studies, *Bifidobacterium* and *Lactobacillus acidophilus* have demonstrated benefits for UC patients, including reduced rectal bleeding symptoms, improved endoscopic scores, and better redox statuses [44,45,46]. The administration of *Lactobacillus reuteri* enemas to children with ulcerative proctitis effectively improved their clinical scores [47]; however, some studies have not observed significant differences with the supplementation of the same species, suggesting that the combined use of multiple strains may be more effective. The De Simone formulation, consisting of eight lactic-acid-producing species, has been extensively studied and shown to provide relief for pediatric and adult UC patients [35,48]. Another effective probiotic mixture is VSL#3, which includes *Lactobacillus*, *Bifidobacterium*, and *Streptococcus thermophilus* [49]. It has demonstrated efficacy in mouse models of UC and mild to moderate UC patients, reducing rectal bleeding, inflammatory markers, and improving mucosal antioxidant capacity. The alleviating effect of probiotics on IBD has been summarized in Appendix A. It is important to note that most patients in clinical settings receive anti-inflammatory drugs as part of their routine care, and investigating the synergistic effects between conventional drugs and probiotics is necessary. Combining VSL#3 with probiotics has shown reduced rectal bleeding frequency [50]. Dual treatment with probiotic mixtures and mesalazine has resulted in shorter recovery times and improved endoscopic images in UC patients [51], whereas some studies have shown that probiotics (*E. coli Nissle 1917* and VSL#3) have no significant therapeutic effect on UC patients [48,50,52]. The controversial results might be due to the differences in trial design, evaluation criteria, treatment duration, research scale, and patient characteristics (such as age, disease development stage, geographical location, and intervention type) [53]. Therefore, it is necessary to carry out a unified design experiment on a larger patient group to correctly evaluate the beneficial effects of probiotics on UC; however, it is generally believed that a longer treatment with the combination of *VSL#3* and *lactobacillus* after surgery may have a better therapeutic effect on UC.

### 3.2. Effect of Probiotics on Alleviating CD

The effectiveness of probiotics in Crohn’s disease (CD) treatment remains disputed. Some studies have shown positive outcomes, such as reduced colonic edema and improved histological scores in CD mice treated with *Bifidobacterium bifidum* in addition to the relief of symptoms in children with CD treated with *Lactobacillus rhamnosus* and *Saccharomyces boulardii* [54,55]; however, other studies have not found significant benefits with the use of probiotics in CD patients, including the effectiveness of *Lactobacillus rhamnosus* and VSL#3 [34,56,57]. Overall, probiotics are not recommended for the treatment of CD patients based on current evidence [47].

Given that IBD is a multifactorial disease, it is not reasonable to use the same probiotic species for all patients to achieve the same efficacy. Personalized medicine should be considered in future studies, taking into account factors such as IBD subtype, the location of the pathology, disease severity, the composition of the patient’s microbiota, and environmental as well as genetic background, to determine the appropriate bacterial strains and doses for individual patients.

## 4. Probiotics Inhibit Intestinal Oxidative Damage in Weaning Piglets

In pig farming, oxidative stress is one of the major causes of disease. The intestine is a main target of ROS attack, which easily leads to intestinal inflammation, barrier disruption, diarrhea, and microbial disorders, ultimately resulting in reduced feed intake and slow weight gain, severely compromising farming benefits. Studies have shown that MDA, protein hydroxyl, and ROS are significantly increased in the liver, intestine, and blood of weaned piglets [58], whereas the activities of GSH-PX and SOD are significantly inhibited [59]. Intestinal oxidative stress induces microbiome dysregulation, leading to post-weaning diarrhea (PWD), and intestinal infections are a major problem facing pig production [9].

Given its antioxidant properties, various strains of *Lactobacillus* have been reported as potentially being able to alleviate piglets’ post-weaning diarrhea. Research shows that an increased mean daily weight gain and a significant decrease in the rate of diarrhea were observed when LPS-challenged piglets were fed *Lactobacillus salivarius*, and also accompanied by decreased levels of proinflammatory mediators (IL-6 β, TNF- α, IL-2, and IFN- γ) in serum and mesenteric lymph nodes [9]. Similarly, the alleviating effect of *Lactobacillus gasseri, Lactobacillus reuteri,* and *Lactobacillus acidophilus* on diarrhea in piglets was demonstrated in several reports, which was associated with a reduction in enterotoxigenic *Escherichia coli* (ETEC) adhesion [11,60]. In addition, *Lactobacillus acidophilus* and *Lactobacillus casei* alleviated the severity of PWD by decreasing systemic immune responses and intestinal oxidative stress [11].

The intestine is the crucial place of nutrient digestion and absorption; thus, villus health remarkably affects the growth conditions of livestock, and is usually evaluated through villi height (VH), crypt depth (CD), and the villus-height-to-crypt-depth ratio (VCR) [49], whereas the shortening of intestinal villi and increased crypt depth via oxidative stress impede nutrient absorption [61]. On the other hand, intestinal nutrient absorption is mainly performed in a transmembrane or paracellular manner, closely related to tight junction proteins [62]. Tight junctions mainly include occludin, claudin-1, and ZO-1 proteins, and their dynamic changes have a key role in regulating the intestinal barrier and cell survival [63]. In addition to the function of digesting and absorbing nutrients, the intestinal epithelium also serves as a barrier against harmful antigens and pathogens. Intestinal barrier damage caused by oxidative stress is usually manifested by the disruption of tight junction proteins and the release of diamine oxidase (DAO) into the serum [64].

Interestingly, research showed piglets fed with *Lactobacillus plantarum* have higher VH as well as VCR and lower CD [65]. Additionally, dietary supplementation with *Lactobacillus delbrueckii* successfully reversed the LPS-induced increase in serum DAO and intestinal CD, as well as raised occludin, ZO-1, and Claudin-1 levels in the ileum of piglets [66]. Pretreatment with *L. salivarius* could stimulate the expression levels of SOD, GSH-PX4, and CAT in the intestine of weaned piglets, but inhibited the immune response and oxidative stress caused by LPS infection, thus restoring the intestinal integrity of the weaned piglets [9]. Furthermore, *Lactobacillus delbrueckii* supplementation alleviated an LPS-induced increase in MDA in serum but decreased jejunal mucosa 8-hydroxy-2-deoxyguanosine levels of piglets [67]. These results suggest that *Lactobacillus* maintains the intestinal epithelial barrier integrity by reducing oxidative stress. Other similar reports are summarized in Appendix A.

## 5. Potential Signaling Pathways Underlying the Antioxidant Actions of Probiotics

Nuclear factor erythroid 2–related factor 2 (Nrf2) is a member of the cap‘n’collar transcription factor family and consists of seven NEH domains. Currently, Nrf2 has emerged as a well established ubiquitin-dependent signaling system in response to OS [68]. High levels of ROS stimulate the separation of Nrf2 from its constitutive inhibitor: Keap1. Subsequently, Nrf2 enters the nucleus and binds to antioxidant response element (ARE) sequences, initiating the transcription of antioxidant genes such as NQO1, GST, HMOX1, GCL, and GSH (Figure 2). A large body of investigation indicates that Nrf2 activation could inhibit OS and inflammation, thereby preventing UC [69]. The activation of the Nrf2 system by probiotics in a host is believed to be one of the important mechanisms through which they exert antioxidant properties. In an in vivo UC rat model, *Lactobacillus delbrueckii* and *Lactobacillus fermentum* play a protective role by upregulating the Nrf2/Ho-1 pathway [70]. Similarly, *Lactobacillus helveticus* has also been shown to activate the Nrf2 pathway, relieving intestinal oxidative stress in mouse models [71]. Furthermore, the remission of LPS-induced intestinal injury by *Bacillus coagulans TL3* is also related to Nrf2 signaling activation [72]. In vitro, the activation of the Nrf2 pathway has also been shown to mediate the antioxidant effect of *Bifidobacterium* infantis, *Clostridium butyricum*, and *Lactobacillus casei Shirota* in intestinal injury [73,74,75]. The activation of toll-like receptors (TLRs) has been reported to stimulate Nrf2-ARE signaling and HO-1, both in vivo and in vitro [76]. Additionally, numerous studies have reported that the stimulation of TLR-Nrf2 signaling by *Lactobacilli* might also be responsible for their antioxidant benefits in the piglets’ guts [77,78].

Nuclear factor kappa-B (NF-κB) is another transcription factor that accounts for the redox mechanism of probiotics. It has been found that NF-κB has a close relationship with Nrf2. Specifically, the loss of Nrf2 can increase NF-κB activity, leading to more serious inflammation, and NF-κB activation can also mediate the transcription of Nrf2 [50]. Lactobacillus species, such as *Lactobacillus johnsonii L531*, *Lactobacillus reuteri*, *Lactobacillus brevis*, and *Lactobacillus fermentans*, have been shown to activate the NF-κB pathway, relieving intestinal OS and inflammation in rats [50,79,80]. Additionally, the artificial modification of *Lactobacillus* expressing SOD showed similar results [81]. Moreover, *Bifidobacterium* downregulates ROS and inhibits NF-κB pathways to modulate the intestinal immune system and protect the intestinal epithelium, as has been addressed in detail [82].

Silent information regulator factor 2-related enzymes (SIRTs) are highly conserved NAD^+^-dependent class III histone deacetylases. Currently, there are seven recognized members in the human SIRT family: SIRT1 to SIRT7.

The crosstalk between SIRT1 and Nrf2/ARE plays a key role in antioxidant defense. In brief, SIRT1 prompts the nuclear translocation of Nrf2, thereby upregulating the expression of antioxidant proteins and phase II detoxification enzymes [83]. In rats with aging-induced colitis, *Lactobacillus C29* treatment decreased the plasma levels of ROS, malondialdehyde (MDA), and C-reactive protein, while increasing SIRT1 expression [84]. Furthermore, the alleviation of high-fat-diet-induced UC by *B. longum* and *L. plantarum* is also associated with the activation of SIRT1 [85]. It has also been shown that activated SIRT2 can deacetylate the forkhead box proteins (FOXO1a and FOXO3a) to increase the expression of FoxO-dependent antioxidant enzymes [86]. The activation of manganese superoxide dismutase (Mn-SOD)/SOD2 by *B. longum* and *L. acidophilus* to reduce cellular ROS levels mediated by SIRT2 has been reported [87].

Mitogen-activated protein kinases (MAPKs) pertain to the serine/threonine kinase family and participate in numerous biological processes, including cell inflammation, antioxidation, and cell death [88]. As of now, three MAPK types have been discovered: ERK, JNK, and p38 MAPK. It is generally accepted that Nrf2 activation mediated by ERK1/2, JNK, and p38 accounts for the expression of phase II detoxifying enzymes [52,89]. *Lactobacillus rhamnosus GG* prevents the H2O2-induced disruption of tight junctions in the human intestinal epithelium, which may be mediated through ERK1/2 [90]. Additionally, both heat-killed and active *Lactobacillus brevis* effectively ameliorated subtotal duodenal and colonic injury caused by mercury poisoning or DSS by blocking oxidative stress and inflammation through a p38-MAPK-mediated pathway [91,92].

## 6. Summary and Outlook

Taken together, IBD and post-weaning diarrhea are complex and multifactorial diseases with unclear direct causes and pathological mechanisms; however, the significant role of oxidative stress in their pathogenesis has been widely recognized. Currently, anti-inflammation and anti-oxidation are important treatment targets for IBD and post-weaning diarrhea.

Studies have shown that probiotics, particularly *Lactobacillus* and *Bifidobacteria*, possess antioxidant properties in mammals. Therefore, providing probiotics appears to be a promising strategy for IBD treatment. Probiotics may improve various pathological aspects of IBD, with mixed-species formulations being more effective than single-species ones. VSL#3 and the De Simone formulation have emerged as the most effective microbial agents for treating UC; however, probiotic formulations seem to be less effective in treating CD compared to UC, potentially due to differences in inflammation location, disease severity, and duration. Similarly, incorporating probiotics into the feed of weaned piglets can significantly reduce diarrhea, intestinal inflammation, and barrier disruption. The antioxidant properties of probiotics mainly involve scavenging free radicals, chelating metal ions, modulating antioxidant enzyme expression, and influencing gut microbiota. At the molecular level, probiotics can impact signaling pathways, such as Nrf-2, NF-κB, MAPK, and SIRTs, to exert antioxidant effects.

It should be noted that animal IBD models do not fully replicate the human immunological profile, particularly in multifactorial diseases. Therefore, the efficacy of probiotics should be tested in various models. Additionally, the lack of standardized evaluation criteria for antioxidant capacity hampers the comparison of results between studies. Moreover, the appropriate dosage, species type, mixing ratio, and treatment duration of probiotic preparations have yet to be determined. Furthermore, chemical drugs and surgical treatments remain preferred in current clinical practices, so the combination therapy of conventional treatment approaches with probiotics should be further investigated. These aforementioned issues should be the focus of future studies.

## Figures and Tables

**Figure 1 antioxidants-12-01342-f001:**
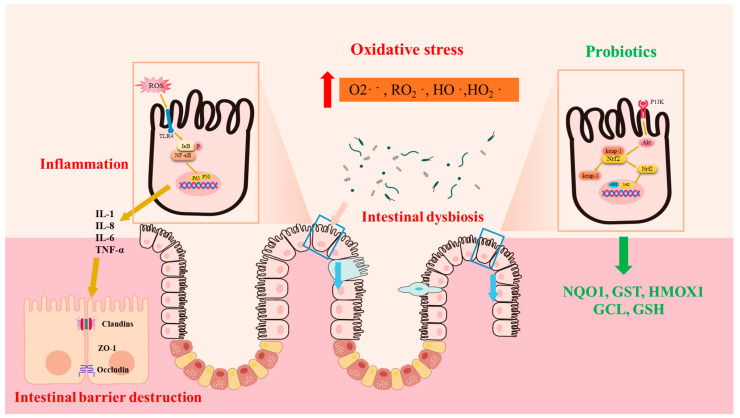
Oxidative stress is associated with the pathogenesis of IBD. Oxidative stress leads to the overexpression of inflammatory factors, which destroy the intestinal barrier and cause opportunistic pathogens to invade the mucosa, exacerbating the vicious cycle. Probiotics stimulate the expression of antioxidant enzymes and relieve IBD by activating the Nrf2 signaling pathway. Superoxide (O_2_·^−^), peroxy radical (RO_2_·), hydroxyl radical (HO·), hydroperoxy radical (HO_2_·), NAD(P)H dehydrogenase quinone 1 (NQO1), glutathione S-transferase (GST), heme oxygenase 1 (HMOX1), glutamate cysteine ligase (GCL), and glutathione (GSH).

**Figure 2 antioxidants-12-01342-f002:**
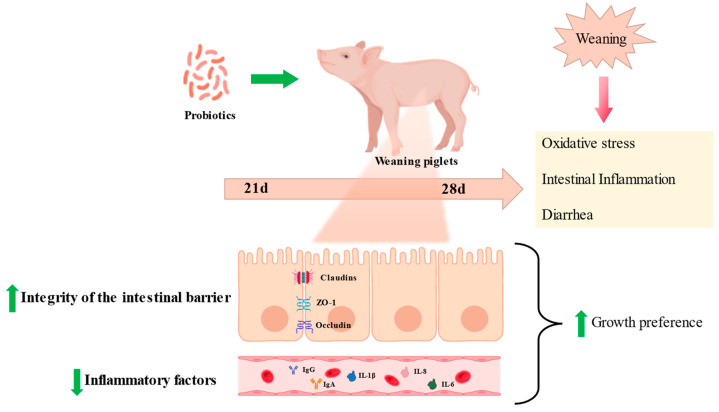
Probiotics relieve weaning stress in piglets. Probiotics can relieve intestinal inflammation, protect intestinal morphology, and reduce the diarrhea rate caused by oxidative stress in weaned piglets.

## Data Availability

Not applicable.

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
