# Peer review of "Exploring the Benefits of Probiotics in Gut Inflammation and Diarrhea—From an Antioxidant Perspective"

_antioxidants, 2023, doi:10.3390/antiox12071342_

Round 1
Reviewer 1 Report
General points about the review: The review brings interesting information regarding the relationship between oxidative stress and inflammatory bowel disease, additionally, authors discussed the potential of probiotics in inflammatory bowel disease treatment and the application of probiotics in weaned piglets. The review is very well written, with sufficient information in the different sections. However, I kindly pointed in this letter some issues that must be reviewed and fixed by the authors, or at least they should explain them better.
Specific considerations:
Title: I believe the species in question (swine) should already be clearly written in the title.
L8 and L9: Should these lines be together?
L13 and L14: Should these lines be together?
Abstract: Please make it clear from the beginning that you are talking about swine.
L32: Should “Lactobacillus” be with capital letter?
L47 and L53: “people” “smoking”? This review is about swine. Should you focus on this, instead of humans? If this investigation wants to infer on the inflammatory bowel disease in humans using swine as a model, please make it clear from the beginning. The review is a mixture of indications on humans, swine and even rats.
L99: Use comma, instead of slash.
L136: Is “Saccharomyces cerevisiae” an opportunistic pathogenic bacterium?
L196: Should this species be in italic? Same L198. Check throughout this review the names that should be in italic.
L207: It has been…
L218: Insert space after ‘frequency’.
L244: “have been could” requires revision.
L249: “and” should not be in italic.
L251: “Escherichia coli” should be in italic.
L256: Was “VCR” previously described?
L266: This sentence requires revision.
L267: You used abbreviation for “crypt depth”, but now wrote as full again. Please be consistent. Check all abbreviations throughout.
Table 2: I believe there should be a better word instead of “species”. Species refers to swine. But what is described in this table is more like development stage.
Table 2: Use italic when appropriate. Describe the abbreviations in footnote.
L285: Abbreviation already described. Check all abbreviations. Spell it out first time mentioned and then only abbreviations onwards. Check throughout.
Figure 2: Place it correctly in the text.
L297: Check italic. Check it throughout the review.
L314, L335, L340: Insert space appropriately.
L343, L346: post.
References: Insert references according to Journal’s guidelines.
Kind regards.
Only minor editing of English language required. Few suggestions were given.
Author Response
Dear reviewer,
Thank you very much for your effort on our manuscript (entitled " Exploring the Benefits of Probiotics in the Gut Inflammation and Diarrhea—From the Antioxidant Perspective"). We have addressed your comments point-by-point and hope the revised version is now acceptable for publication in " Antioxidants". For your information, we have revised the manuscript accordingly as follows:
L8 and L9: Should these lines be together?
Answer: We have made the modifications in line 8-9.
L13 and L14: Should these lines be together?
Answer: Yes, we have made the modifications in line 13-14.
Abstract: Please make it clear from the beginning that you are talking about swine.
Answer: We highlighted the content of "weaned piglets" in the line 28-29, 38-39.
L32: Should “Lactobacillus” be with capital letter?
Answer: Yes, we have made the modifications in line 32.
L47 and L53: “people” “smoking”? This review is about swine. Should you focus on this, instead of humans? If this investigation wants to infer on the inflammatory bowel disease in humans using swine as a model, please make it clear from the beginning. The review is a mixture of indications on humans, swine and even rats.
Answer: We explained the reason why we chose pig and mouse models in line 86-94.
In clinical research, difficulty in sampling, environmental limitations, and ethical ethics often hinder direct research on human diseases. For a long time, small rodents have been important model animals for basic medical research, making significant contributions to the understanding of the pathogenesis and treatment of human diseases. However, pigs are more like humans in physiology and anatomy than rodents, making them a potential choice for simulating human diseases. At present, there are pig models for cardiovascular diseases, metabolic disorders and neurological diseases a, which provide considerable support for the analysis and treatment of human diseases. Therefore, we take the intestinal inflammation of weaned piglets as an example to discuss the following problems.
L99: Use comma, instead of slash.
Answer: We corrected it in line 119.
L136: Is “Saccharomyces cerevisiae” an opportunistic pathogenic bacterium?
Answer: We deleted this part.
L196: Should this species be in italic? Same L198. Check throughout this review the names that should be in italic.
Answer: Yes, these species be in italic. We also checked the full text and adjusted the format of species names.
L207: It has been…
Answer: The alleviating effect of probiotics on IBD has been summarized in Table 1.
L218: Insert space after ‘frequency’.
Answer: we have Inserted space after ‘frequency’ in line 220.
L244: “have been could” requires revision.
Answer: We corrected it in line 263. Given its antioxidant properties, various of Lactobacillus have been reported could alleviate piglets post-weaning diarrhea.
L249: “and” should not be in italic.
Answer: We corrected it in line 268.
L251: “Escherichia coli” should be in italic.
Answer: We corrected it in line 270.
L256: Was “VCR” previously described?
Answer: We corrected it in line 275. Villus height to crypt depth ratio (VCR)
L266: This sentence requires revision.
Answer: We rewrote this sentence in line 285. Interestingly, research showed piglets fed with Lactobacillus plantarum have higher VH and VCR and lower CD.
L267: You used abbreviation for “crypt depth”, but now wrote as full again. Please be consistent. Check all abbreviations throughout.
Answer: We corrected it and check all abbreviations throughout.
Table 2: I believe there should be a better word instead of “species”. Species refers to swine. But what is described in this table is more like development stage.
Answer: We corrected it.
Table 2: Use italic when appropriate. Describe the abbreviations in footnote.
Answer: We have added abbreviations.
L285: Abbreviation already described. Check all abbreviations. Spell it out first time mentioned and then only abbreviations onwards. Check throughout.
Answer: We corrected it in line 300. We have carefully examined the full text.
Figure 2: Place it correctly in the text.
Answer: We adjusted the position of the picture.
L297: Check italic. Check it throughout the review.
Answer: We corrected it and we have carefully examined the full text.
L314, L335, L340: Insert space appropriately.
Answer: We corrected it and we have carefully examined the full text.
L343, L346: post.
Answer: We corrected it
References: Insert references according to Journal’s guidelines.
Answer: We have corrected the reference format according to the magazine's requirements
Reviewer 2 Report
In general, the manuscript has the potential for publication.
Herebelow, some points for improvement are noted.
1. Authors need to explain the methodology through which they selected the references that they reviewed.
2. Authors should move the table to appendices, as they are lengthy (albeit useful) and hinder the flow of reading.
3. Authors need to separate sections 2 and 3 into subsections.
4. Authors need to increase their critical evaluation of the references presented and discussed.
5. Authors need to insert some more recent literature references in this study.
Overall: reevaluation after correction.
Moderate editing of English language required.
Author Response
Dear reviewer,
Thank you very much for your effort on our manuscript (entitled " Exploring the Benefits of Probiotics in the Gut Inflammation and Diarrhea—From the Antioxidant Perspective"). We have addressed your comments point-by-point and hope the revised version is now acceptable for publication in " Antioxidants". For your information, we have revised the manuscript accordingly as follows:
- Authors need to explain the methodology through which they selected the references that they reviewed.
Answer: We first focus on the timeliness of references and try to select articles published in the past decade as much as possible. In addition, the references we have selected are representative, and these reports come from various countries around the world. When quoting, we identify and comment on biased and biased views. Finally, we try to select research articles from publishing institutions with good academic reputation to ensure their authority.
- Authors should move the table to appendices, as they are lengthy (albeit useful) and hinder the flow of reading.
Answer: We adjusted the position of the tables.
- Authors need to separate sections 2 and 3 into subsections.
Answer: We have separate sections 2 and 3 into subsections according to the suggestion.
- Authors need to increase their critical evaluation of the references presented and discussed.
Answer: We have added this section. Whereas, some studies have shown that probiotics have no significant therapeutic effect on UC patients. The controversial results might be due to the differences in trial design, evaluation criteria, treatment duration, research scale and patient characteristics (such as age, disease development stage, geographical location and intervention type). Therefore, it is necessary to carry out a unified design experiment on a larger patient group to correctly evaluate the beneficial effects of probiotics on UC. However, it is generally believed that the longer treatment with the combination of VSL # 3 and Lactobacillus after surgery may have a better therapeutic effect on UC.
- Authors need to insert some more recent literature references in this study.
Answer: We followed the suggestions and referred to some of the latest research.
Reviewer 3 Report
Right from the first paragraph, i.e. the abstract, there is a jumble and confusion of information concerning IBD in humans (patients) and in weaned piglets. It is hard to keep up with the keynote. Is it about the piglets themselves or are weaned piglets a model for humans? This needs to be greatly revised so that the reader has a sense of the authors' clear purpose.
Table 1: there is no information how the IBD was induced in those in vitro models. Add that in separate column. Under the table provide information about all abbreviations used (same table 2).
---------------------------------------
The authors tried to take a deeper look at mechanism related to oxidative stress and IBD. Then, in details they provided current information for probiotics as a possible cure for that illness. As a research model the weaned piglets were chosen.
Although the topic is not entirely original as the literature provides tons of knowledge about ROS and some diseases, there is still a gap about specific mechanisms how reactive oxygen species damage the body systems and organs.
I suppose that the literature is quite rich in knowledge about the topic but the authors gave us the opportunity to see more details because more relevant analyses can be done using animal models. Therefore some new information has been added by that work.
Right from the first paragraph, i.e. the abstract, there is a jumble and confusion of information concerning IBD in humans (patients) and in weaned piglets. It is hard to keep up with the keynote. Is it about the piglets themselves or are weaned piglets a model for humans? This needs to be greatly revised so that the reader has a sense of the authors' clear purpose.
The conclusions are right and justified by the main text.
The references were used in an appropriate manner.
Author Response
Dear reviewer,
Thank you very much for your effort on our manuscript (entitled " Exploring the Benefits of Probiotics in the Gut Inflammation and Diarrhea—From the Antioxidant Perspective"). We have addressed your comments point-by-point and hope the revised version is now acceptable for publication in " Antioxidants". For your information, we have revised the manuscript accordingly as follows:
Right from the first paragraph, i.e. the abstract, there is a jumble and confusion of information concerning IBD in humans (patients) and in weaned piglets. It is hard to keep up with the keynote. Is it about the piglets themselves or are weaned piglets a model for humans? This needs to be greatly revised so that the reader has a sense of the authors' clear purpose.
Answer: Weaned piglets a model for humans, which is used to illustrate the human situation.
Table 1: there is no information how the IBD was induced in those in vitro models. Add that in separate column. Under the table provide information about all abbreviations used (same table 2).
Answer: We added the necessary information.
The authors tried to take a deeper look at mechanism related to oxidative stress and IBD. Then, in details they provided current information for probiotics as a possible cure for that illness. As a research model the weaned piglets were chosen.
Answer: Yes, your comment is consistent with what we are trying to convey.
Although the topic is not entirely original as the literature provides tons of knowledge about ROS and some diseases, there is still a gap about specific mechanisms how reactive oxygen species damage the body systems and organs.
Answer: We mainly focus on the damage of intestine caused by ROS. Specifically, ROS overload leads to the activation of NF-κB pathways and the overexpression of inflammatory factors. Subsequently, the destruction of tight junction leads to the destruction of intestinal barrier and morphology. In addition, the disruption of the intestinal barrier allows opportunistic pathogenic bacteria to invade the mucosa, further exacerbating the inflammatory response.
I suppose that the literature is quite rich in knowledge about the topic but the authors gave us the opportunity to see more details because more relevant analyses can be done using animal models. Therefore some new information has been added by that work.
Answer: Thanks for your recognition of our work.
The conclusions are right and justified by the main text.
Answer: Thanks for your recognition of our work.
The references were used in an appropriate manner.
Answer: Thanks for your recognition of our work.
Round 2
Reviewer 2 Report
Answer: We first focus on the timeliness of references and try to select articles published in the past decade as much as possible. In addition, the references we have selected are representative, and these reports come from various countries around the world. When quoting, we identify and comment on biased and biased views. Finally, we try to select research articles from publishing institutions with good academic reputation to ensure their authority.
Can the authors please add their response into the manuscript, to describe their methodology?
Minor editing of English language required.
Author Response
Can the authors please add their response into the manuscript, to describe their methodology?
Answer: We have added the following content in the introduce section, in line 98-105.
We conducted a thorough search using PubMed, Medline, and Web of Science databases from 2010 to 2023, and found a total of 41 papers to include in this review. These papers included in vitro cell tests, small rodents (mainly induced with dextran sodium sulphate or 2,4,6-trinitrobenzenesulfonic acid), and piglets (induced with weaning stress) as IBD models. In addition, we also searched for double-blind, placebo-controlled trials of adults and children with active or quiescent CD or UC within the past 20 years. This review provides valuable insights into the potential of probiotics, particularly lactobacillus, in alleviating IBD symptoms.
